# Ionic EAP Actuators with Electrodes Based on Carbon Nanomaterials

**DOI:** 10.3390/polym13234137

**Published:** 2021-11-26

**Authors:** Nikolay I. Alekseyev, Ivan K. Khmelnitskiy, Vagarshak M. Aivazyan, Anton P. Broyko, Andrey V. Korlyakov, Victor V. Luchinin

**Affiliations:** Department of Micro- and Nanoelectronics, Faculty of Electronics, Saint Petersburg Electrotechnical University “LETI”, 197376 Saint Petersburg, Russia; khmelnitskiy@gmail.com (I.K.K.); aivazyanvm@mail.ru (V.M.A.); broyko@gmail.com (A.P.B.); akorl@yandex.ru (A.V.K.); cmid_leti@mail.ru (V.V.L.)

**Keywords:** IPMC actuator, IPGC actuator, conductive polymers, flexible electrodes, carbon nanotubes, graphene, graphene oxide, reduced graphene oxide, graphdiyne, hot pressing

## Abstract

Flexible polymer-based actuators, often also called artificial muscles, are an essential part of biomimetic systems that mimic the movement principles of animal world creatures. The most used electrode material to force the actuator move is an ensemble of noble metal nanoparticles in the electroactive polymer surface. Noble metal electrodes have enough electrical conductivity and elasticity and are not subjected to oxidation. However, high cost of such electrodes and their tendency to cracking dictate the need for searching other materials, primarily carbon ones. The review considers several options for this search. For example, carbon nanotubes and graphene have excellent properties at the level of a single individually taken nanotube or graphene sheet. However, conservation of these properties in structurally imperfect film electrodes requires a separate study. In addition, there are problems of compatibility of such electrodes with the polymers that requires cumbersome technologies, e.g., hot pressing, which complicates the production of the actuator as a whole. The review concerns the technology options of manufacturing actuators and the results obtained on their basis, both including hot pressing and avoiding this procedure. In particular, the required level of the graphene oxide reduction in hydrazine provides sufficient adhesion at rather high electrical conductivity of the graphene film. The ability to simultaneous achieving these properties is a nontrivial result, providing the same level of actuation as with expensive noble metal electrodes. Actuators that additionally require greater lifetime resource should be obtained in other ways. Among them are using the graphdiyne electrodes and laser processing of the graphene electrodes.

## 1. Introduction

The use of carbon materials for the both electrodes and the electroactive membrane materials in actuators and other biomimetic devices is an intensively developing area in the progress of these devices. However, there are not many original works and reviews on this subject.

This is due to a number of serious problems in using carbon materials. In their original form, these materials are essentially hydrophobic and therefore are poorly combined with the water base of electrolytes, which is the most effective environment for the actuators, and with water-filled membranes. On the other hand, in the process of functionalization, the conductive properties of carbon materials deteriorate notably. The existing methods of connecting electrically conductive membranes with ionic electroactive polymers (EAPs) are still either insufficiently effective or too expensive, and no effective solution has yet been found.

This review partially overlaps the existing and well-known reviews on carbon materials as applied to the actuators [1,2,3] and concentrates in some more details on the results obtained on this subject in our previous works [4,5].

In the classical and the most advanced version, the electrode composite for the electrochemical actuator is formed on the basis of noble metal particles, impregnated into an ionic EAP; this polymer is filled with electrolyte. The entire composite structure of the actuator is called an ionic polymer-metal composite (IPMC) [1,2].

The IPMC actuator principle of operation is based on transport processes occurring inside a polymer membrane. During saturation with a liquid, a dry polymer is structured in such a way that the polymer chain hydrophilic ends are oriented towards the membrane pores filled with an ionizing solvent. The action of the electric field induced by the voltage applied to the electrodes leads to the charged liquid movement in the membrane along the system of through pores due to the processes of diffusion and electroosmosis. The arising electroosmotic fluid flow causes an increase in pressure at one electrode and a decrease near the other electrode. The pressure drop leads to the actuator deformation [6].

The development of IPMC transducers is strongly restrained by the noble metal price (as a rule, platinum or palladium) and electrode brittleness, resulting from conflicting requirements to the electrodes: flexibility, high conductivity, and a significant service lifetime at the same time. Besides that, the procedure of chemical coating of the Nafion membrane with electrode layers (the most common technology) requires a great deal of effort and time.

For these reasons, an intensive search is under way for cheap electrode materials that possess the above combination of properties. This combination can be typical of carbon materials, primarily graphene and carbon nanotubes (CNTs), provided that they are produced in a fairly cheap way. For example, we can hardly talk about the prospects of mechanically exfoliated graphene, a standard material that demonstrates the unique electronic properties of graphene [7].

The use of carbon materials in ionic EAP is not limited to replacing the noble metals of the electrodes with these materials. Another direction may be the ability to provide a high ionic conductivity of the membrane, which limits the operation of the actuator at high frequencies. Further, the attention will be focused on the latest results of the nanocarbon materials research: nanotubes, graphene, and graphdiyne as electrode materials for actuators.

Although many different actuators (acoustic, electromechanical, thermal, etc.) have been developed on the basis of carbon nanomaterials, we will restrict ourselves to ionic EAP actuators, whose standard components are the polymer and the electrolyte that fills it. Such actuators can be used in robotic systems as artificial muscles [1] as well as in microfluidic pumps as drives for drug delivery [8].

## 2. Features of Graphene as a Basic Material of the Nanocarbon Materials

The properties of graphene are the basis for understanding the reasons for interest in the entire family of carbon nanomaterials intensively studied at present, such as the graphene itself, CNTs, nanoribbons, fullerenes (to some extent), and some others.

The key properties that determine the prospects for using the graphene in actuators are high-charge carrier mobility and exceptional elastic properties. Measurements carried out in 2005 through the 2010s [7,9] indicate the mobility value of 2 × 10^5^ cm^2^/(V∙s) for a free mechanically exfoliated graphene monolayer.

Indeed, this huge value is only realized as high conductivity in the case of sufficiently high carrier concentration. Due to the specificity of graphene as a gapless semiconductor, this concentration is formally equal to zero for free graphene. By applying a gate voltage to graphene, it is easy to achieve the concentration as high as 10^12^ cm^−2^, but the mobility decreases noticeable due to the influence of the substrate.

When creating such a potentially mass product as actuators, the graphene is usually represented as chemically exfoliated graphene, which is a product of a complex graphite exfoliation technology with strong acids. Such graphene always contains a significant number of grafted functional groups, some of which do not leave after material recovery of any kind.

Thus, in the electrodes of actuators, graphene is always unintentionally doped, but the carrier mobility in it is quite low due to a significant number of reduction defects and residual functional groups. In addition, in a polycrystal electrode composed of a large number of graphene flakes, the actual conductivity is determined by the contacts between the flakes. Therefore, the feasibility of high mobility and, consequently, conductivity in practical graphene is relatively low.

The same applies to elastic properties. So far, we cannot talk about a continuous graphene monolayer, which would cover the actuator surface and which could be rolled up into a tube without damage. The ability of a real polycrystalline graphene film to return to its original position at the level of individual graphene flakes should be evaluated in each case experimentally.

Another property of the graphene (and, in part, CNTs), important for actuators, is the quantum capacity and the associated quantum elongation [10]. This effect is caused by the low density of states (DOS) in graphene at energies ε close to zero, as the DOS value is proportionate to ε [11]. In such a system, the charge process of the actuator as a capacitor is regulated by the need to preliminarily increase the DOS in the electrodes to a value capable of accepting a given amount of charge. Carrying out such a “zero cycle” means an additional capacity turned in series with the own capacity.

This quantum capacity, which “swallows” a huge part of the real capacity of carbon nanomaterial, is, in particular, the reason for the unattainability of the potentially high capacity of graphene (about 2500 cm^2^/g) in supercapacitors by almost an order of magnitude [12].

An increase in the Fermi energy and the DOS required to accept a charge is achieved not only by the growth of this charge itself but also by the elongation of the electrodes (quantum mechanical elongation).

## 3. Actuators with Carbon Nanotube Electrodes

The peak of interest in the properties of the first of the listed materials (CNTs) was in the late 1990s–2000s. The primary motivation for using CNTs in actuator electrodes was the same reason as the apparent prospects of CNTs in the hydrogen accumulation problem—a huge specific surface area and electronic conductivity at one time. The other advantages were flexibility and electrochemical stability. However, it quickly became apparent clear that the hydrogen sorption capacity of all real nanotube materials is insufficient.

On the other hand, the technique of obtaining thin nanotube paper (buckypaper, the adjective “bucky” is a reminder of buckminster-fullerene as the first historical name of the fullerene molecule), suitable for studying nanotubes in applied problems, was developed. As applied to actuators, such paper turned out to be a good substitute for traditional metal-film noble metal electrodes. Another option was composite thin films of gel-type nanotubes with a polymer or with an ionic liquid (IL) or with the both of these components (bucky-gel).

The diagram of how a bucky-gel electrode actuator responses to a meander voltage applied across the electrodes is shown in Figure 1. The unperturbed actuator membrane length from the attachment line *L*, the displacement amplitude *D*, the curvature radius *R*, and the outer side stretching (or the inner side compression) Δ*L* with the membrane thickness *d* and at small *D*/*L* ratio are determined by the almost obvious formulas:$$R=L^{2}/2D,\;\Delta L=dD/L.$$

Except the stress and the actuator displacement amplitude (or the equivalent maximum tension value ε_max_), the principal parameter of the actuator is the maximum mechanical stress σ generated during actuation and recalculated from ε_max_ and the Young’s modulus Y by the Hooke’s law: σ = ε_max_Y.

The most valuable and hard-to-obtain kind of the CNTs are single-walled carbon nanotubes (SWCNTs) generated by CVD technique or in arc with a metallized graphite anode. SWCNTs separated by subsequent chemical processing play the role of the CNT property demonstrator in the CNT research. This role is similar to that of mechanically exfoliated graphene in the graphene science. Various types of actuators have been developed on the basis of SWCNTs. In particular, the first electrochemical SWCNT-based actuator was created in 1999 by the authors of [13].

The strips of the nanotube paper, cut by the shape, were adhesively attached to the opposite sides of a double-sided adhesive tape and a high voltage was applied to them. The principal mechanism of the structure actuation, according to the authors, was the electronic mechanism based on the electron injection into one electrode and the hole injection into the other one. As shown in quantum-chemical calculations, the electron injection is accompanied by significant tube stretching [14]. On the opposite electrode, on the contrary, holes are injected and the electrode must be compressed.

This mechanism obviously works in the absence of ion movement as well. In fact, such electrostatic effect is very small regardless of the sign of the charge: the strip expands to a much greater extent than stretches.

In the case of a SWCNT electrode in an electronic type actuator described in [13], the effect arose when the field between the electrodes reached 4 V/nm.

Another mechanism of actuation, different from electronic but similar to it, is ionic mechanism that is associated with the movement of the cations and anions. First of all, more mobile cations entrain water in a water-filled polymer. Owing to that, the stretching and the compression of the electrode occurs as a mechanic-hydrodynamic effect.

In the case of IL, the stretching of one electrode and the compression of the other one occur due to significantly different sizes of the cation and anion. In addition, in the case of IL, near-electrode layers with different charge signs should be formed near the electrodes; thus, a capacitor arises. When it is charged/discharged by the electrolyte ions, both quantum elongation and electrostatic actuation as a sequence of the charge injection (Figure 2) should occur.

Indeed, the electron-ion system of an individual nanotube at each of the electrodes, together with the adjacent electrolyte ions, forms an open quantum-mechanical system (Figure 2a). Therefore, the electrolyte cation movement towards this electrode and then inside it should be compensated by the injection of electrons into the nanotube. It is accompanied by stretching.

In addition, in the quasineutral ensemble “the nanotube electrons—the C^+^ ions in it—the cations”, it is the cations that face the electrolyte and form a capacitor with the anions at the opposite electrode.

In what follows, we will consider only ionic actuators, whose displacement is caused by the ion migration in the electrolyte layer that fills the framework polymer (not necessarily electroactive) and then inside the electrode layers. In this case, a significant bending of the membrane occurs already at a low voltage of 1–5 V applied to the electrodes.

When studying the ionic polymer composites with nanotube electrodes, the initial material of such electrodes was a SWCNT gel film with IL (SWCNT/IL-gel) [15].

It was found this way that only SWCNTs could form a good physical gel with an IL among all carbon materials (graphene, fullerene C_60_, and CNTs). This is due to the interaction between IL cations and π-electrons of nanotubes. This is the reason why graphene, which have recently been produced in large quantities and by cheaper methods than nanotubes, have in no way ousted them from the actuator topic.

In [15], the electrode composite was obtained by heating the SWCNT/IL-gel in the same material as the polymer skeleton of an actuator—a mixture of poly(vinylidene fluoride-co-hexafluoropropylene) (PVDF-HFP) and 4-methyl-2-pentanone at temperature of 80 °C. It was this material that figured in most of the studies carried out with IL-filled actuators. 1-Ethyl-3-methylimidazolium (EMIM) salts were most often used as ILs. Thus, only the presence of the nanotubes differs the composition of the electrode from that of the polymeric membrane.

In [16] that followed [15], a sufficiently strong response up to 5 mm was obtained to the meander signal at a frequency of 0.1 Hz. Indeed, the actuator strip was very narrow (~1 mm). These figures can be considered as initial ones. The mechanical stress σ developed in the actuator exceeded the stress in the metal-film electrodes to reach ~1%.

In the case of the SWCNT paper, the level of results achieved turned out to be noticeably lower, and this is quite understandable: the absence of a seamless junction between the electrode and electrolyte should lead to a deterioration in ion transport in the both external and internal layers.

Further optimization of the ionic EAP actuators with the SWCNT gel-based electrodes was carried out in the following directions:Variations of the SWCNT concentration and especially the method for this synthesis;Variations of the nanotube mass grinding methods, e.g., the use a planetary mill instead of a ball mill; due to the complexity of the nanotube dispersion preparation, this direction is closely related to the first one;Varying the methods of a joining the electrode layers with the membrane body (e.g., the hot pressing instead of long-term, layer-by-layer casting);The polymer effect study: it is known that the polymer able to give the mechanical integrity to the ribbon reduces the conductivity and capacity of the electrode layer; from this viewpoint, the optimal electrode film should not contain a polymer at all.

The optimization in the first two directions enabled high stretching amplitudes [17] together with high mechanical stress: 2% and 4.7 MPa for the actuator tape of 0.5-mm thick (5-mm long, 4-mm wide) at alternating voltage of 2.5 V amplitude. The maximum bending force was 0.04 N, which is also noticeably higher than in the case of metal-film electrodes.

The way of the SWCNT synthesis turned out to be the most important from the first two factors. The effective SWCNT film with an IL containing no polymer at all was successfully created but by using an extreme type of nanotubes—extra-long tubes obtained in the CVD process with water [18].

Herewith, the Young’s modulus and tensile strength were ~160 and 15 MPa, respectively; the conductivity was 170 S/cm. After the hot pressing of the composite electrode with the IL/PVDF electrolyte layer, the actuator strip could bend almost perpendicularly to its initial position under DC voltage of 3 V. It is also significant that with the frequency increase from 1 to 10 Hz, the displacement amplitude decreased from 5 to 4 mm, i.e., very slightly. In air, the actuator strip could operate in a continuous mode (±1.0 V, 1 Hz) up to 10,000 cycles.

To explain the fast response and significant displacement of the SWCNT/IL actuator electrodes at high frequency and low voltage, the authors of [19] used the electrochemical actuation mechanism, which was described for ion actuators with standard copper-platinum electrodes in the fundamental review [20].

The mechanism consists in the redox reactions on the surface of an electrochemically active electrode, as a result of which a significant additional charge is transferred through the “electrode/electrolyte” interface. Similarly, in narrow spaces between nanotubes, the actuation barrier of reactions is significantly reduced, additional electrochemical currents are formed, and an electrochemical contribution to actuation appears.

The electrochemical component influence can be sharply increased if capacitive materials, such as conductive polymers and metal oxides, are introduced into the electrodes. Thus, in [21], polyaniline (PANI) and carbon black were added to the electrodes to increase the conductivity and capacity. PANI is a conductive polymer with a high specific capacity used not only in ionic EAP actuators but also in supercapacitors. The electrode layers were obtained by mixing the solution of PANI/SWCNT with IL, PVDF-HFP, and gelatin.

When compared in terms of the tension and the mechanical stress produced, the electrode structures “CNT/PANI” (50/10) turned out to be about five times better than the pure CNTs structures. The conductivity was 8.6 S/cm, and the capacity was 0.0852 F/cm^2^; the tension at an amplitude of 2 V and a frequency of 0.1 Hz was 0.88%, and the mechanical stress was 390 MPa. However, the electrodes could only perform effectively at low frequency. The authors saw the reason for this in slower proceeding of the redox reactions, during which the mechanical tension increases, in the presence of polymer.

As already mentioned, SWCNTs, especially ultra-long ones, are very expensive and cannot be considered as a suitable modifier for such widespread use as actuators. Therefore, the possibility of much cheaper multi-walled carbon nanotubes (MWCNTs), which also have their own advantages and disadvantages, was also investigated in parallel.

Whereas in SWCNTs, any chemical modification, which includes carboxylation and aminization, sharply reduces the electrical conductivity, such manipulation in MWCNTs affects the outer wall only, and the high conductivity set by the core of the tube remains. At the same time, it is easier to fix on the tube surface a group that enhances certain properties. Therefore, MWCNTs were under attention in the development of the other applications of nanotubes—double-layer capacitors and sensors.

In addition to the properties enlisted above, the levels of “entanglement” and interaction between individual tubes are completely different in the SWCNT and MWCNT as in actually obtained materials. Thus, SWCNT films have a higher Young’s modulus (695 MPa) than MWCNT films (470 MPa) and, at the same time, relatively higher van der Waals interactions between individual tubes. The denser packing causes the nanotubes to “stick” together and to be packed more tightly in SWCNT nanotube paper than in MWCNT paper. This factor blocks ion transport to the periphery of the actuator and reduces the tension. This is especially true for films containing aligned nanotubes [22].

On the other hand, the disadvantages of MWCNTs in the problem of actuation are as significant as that of SWCNTs. Therefore, in the absence of real ion transfer through the nanotube channels, the effective surface area of the nanotube is small (10–500 m^2^/g). Thus, the solution can be either surface modification or stimulation of the electrochemical component of the actuation.

The first direction, i.e., the surface modification, was developed in 2012 by the authors of [23], who assembled the actuator with electrodes based on modified MWCNTs, more efficient than the actuator with unmodified MWCNTs or SWCNTs.

The conductivity of the actuator with the MWCNT-COOH type electrodes was 4.6 S/cm, and the capacity was 33 F/g; the stretching was at an amplitude of 2 V and a frequency of 0.05 Hz was 0.8%, and the mechanical stress was 190 MPa. The authors attributed this result to the higher BET specific surface area, a larger pore volume, and better wettability of oxygen groups grafted to the MWCNTs.

The second direction mentioned above and intensively developed at this time is the addition of capacitive materials, such as RuO_2_ [24] and MnO_2_ [25], to the electrode gel.

The last paper of this series [26], appearing in 2016, effectively uses a combination of unmodified MWCNTs with graphene. Herewith, the enhancement of the actuation properties turns out to be greater than when summing up the effect of adding each of the components separately (synergistic effect).

The principal practical result of [26] is the stretching as a function of the applied triangular voltage amplitude of 2 V is shown in Figure 3 for the actuator with gel electrode of the “polymer/CNT/EMIM[CF_3_SO_3_]” type. The IL considered here is the CF_3_SO_3_ anion, which proved to be optimal when comparing various ILs: [CF_3_COO] > [BF_4_] > [TFSI] > [CF_3_SO_3_] > [CF_3_BF_3_].

The capacity of the actuator with the maximum tension was 196 F/g and was within the range of 22–200 F/g for different electrode materials with the participation of MnO_2_; for the materials with the MWCNT/IL electrodes, the capacity was much lower: 10–19 F/g.

## 4. Actuators with Graphene Electrodes

Graphene, which has been intensively studied as electrode material for the ionic EAP actuators with some delay to the study of CNTs, has its advantages and disadvantages. First of all, the theoretically maximum specific surface of graphene (2500–2600 m^2^/g) is of the same order of magnitude as that in SWCNTs and much more than in MWCNTs (however, this surface is almost an order of magnitude lower, which is associated with the quantum capacity [10,12]). More significant is the geometric factor: when the solvated ions are introduced into the graphene, the expansion of the flat-flaked structure is much greater than that of nanotubes; in the direction perpendicular to the basal planes, the expansion is up to 700%.

This occurs due to the intercalation of solvated ions (e.g., Li^+^ ions) between the graphene layers followed by the electrolyte penetration [27]. More mobile cationic component of the IL can also partially pass through the slits between the graphene flakes; then, when the field is applied, it pulls the IL behind itself as a whole.

At the same time, graphene is characterized by the problem of “restacking” the graphene layers. This effect is associated with the van der Waals interaction destruction between the adjacent graphene flakes during the stretching and restoration of the interactions in new positions during the compression. As already noted, the second difficulty is the worse graphene dispersion in ILs as compared with that of nanotubes. These reasons force the use of technologies that have no direct analogs when working with nanotube electrodes.

The first flat electrochemical IPMC actuator with hybrid graphene/binder electrode with IL as the electrolyte was likely to be described in [28]. The reduced graphene oxide (RGO) was used as the graphene. The primary product for obtaining the RGO flakes was high-quality pyrolytic graphite, between the layers of which the strongly oxidizing agent molecules were intercalated. The transformation of graphite into graphene oxide (GO) was carried out in [28] and in the subsequent researches by the Hummers’ method with various modifications [29].

The experiment confirmed the possibility of a large change in volume; in the direction perpendicular to the graphene membranes, it reached 100% and was an order of magnitude higher than that in actuators with CNTs. On the other hand, with an increase in the IL mass fraction in the composite electrode from 0 to ~70 mass.%, the specific capacity of the actuator increased from 2 to 60 F/g. At the same time, the stability of the structure remained low. Thus, a composite with an IL mass fraction of 70% withstood no more than 20 cycles at an operating voltage of 2 V and a frequency of 0.01 Hz. The breaking of the electrodes occurred even faster with higher voltage or with an increase in the half-period of actuation. The authors attributed this to insufficiently effective distribution of the binder within the RGO layers.

Low stability of the hybrid electrode from the IL/graphene composite stimulated the development of crosslinked porous electrodes. The CNTs became a natural candidate for such crosslinking. The RGO-based electrode combined with MWCNTs is described in [30].

It is intuitively clear that this combination should block the restacking, as the flakes stitched with transverse “rafters” should return to their places after stretching and compressing of the electrodes. The conductivity, the ion transport, and electrochemistry are also improved.

The RGO/MWCNT hybrid was synthesized via additional reduction of the GO/MWCNT structure in hydrazine, under which the tubes short-circuit the graphene sheets (Figure 4).

The frequency dependence of the actuator displacement amplitude *D* is given in Figure 5a. Figure 5b illustrates the decrease in *D* as compared with the initial amplitude *D*_0_ in the course of operation.

With a voltage amplitude of 2 V and a frequency of 1 Hz, the stretching amplitude is 0.03%. The other integral parameters of the actuator are given in [30]. The conductivity of the structure is 135 S/cm.

Thus, one can see that the actuators with RGO-based electrodes are capable of providing a sufficiently high mechanical force, especially in combination with nanotubes. However, their conductivity and electrochemical activity remain low due to uncontrolled defects, the edges, as well as the gradual degradation of properties.

An important stage on the way of the conductivity increase (and therefore the actuation stability increase) was the paper [31], where the graphene electrode (RGO-based, as before) was modified with silver so that a hybrid electrode RGO/Ag was developed. The production of this composition was achieved by successive reduction of the silver salt and GO.

The RGO was synthesized in the usual way. Then the GO suspension was mixed with the Ag(NH_3_)_2_ solution. The positive Ag(NH_3_)_2_^+^ ions are readily absorbed on the negatively charged GO surfaces and are reduced with hydrazine. This leads to nucleation of the Ag nanoparticles (AgNPs) and their deposition on the GO surfaces. Then, the GO/Ag mixture is left overnight at 95 °C so that the mixture is reduced up to RGO/Ag (Figure 6).

The hybrid electrode was obtained by evaporation of the RGO/Ag suspension onto a glass substrate. The film formed had a conductivity of 900 S/cm, much higher than the value of 50 S/cm, typical of the RGO electrodes. When producing the actuator with the RGO/Ag electrodes, two hybrid electrodes are connected by hot pressing with the electrolytic membrane of PVDF-HFP filled with the traditional IL—1-butyl-3-methylimidazolium tetrafluoroborate (BMIMBF_4_).

As it is seen from Figure 7a, with electrical stimulation by a 1-V meander signal with a frequency of 0.01–10 Hz, the introduction of RGO increases the actuator displacement in comparison with purely silver electrodes. The operating time gain is even more obvious (Figure 7b).

The authors of [31] suppose that the reason for increasing the actuator displacement amplitude consists in the fact that the standard farad capacitance inherent in an electrochemically active metal is added by the graphene quantum capacitance.

The second important effect is that the actuator with combined electrodes successfully “holds” the meander shelf at maximum amplitude (Figure 8a), in contrast to the actuator with silver electrodes, in which the displacement almost immediately decreases (straightening back, Figure 8b). The reasons for this “non-retention” of the initial displacement of the actuator during the half-period of the meander are clear. When the third charged component (Ag^+^) appears in the electrolyte and adds to the IL components (BMIM^+^ и BF_4_^−^), the rapid displacement is determined by the Ag^+^ and BF_4_^−^ ions that form the primary excessive concentrations of the Ag^+^ cations at the cathode and the BF_4_^−^ anions at the anode.

At the background of this distribution, the inertial component of the cations (BMIM^+^), which are larger in size and capable of holding the cathode stretching, almost cannot reach the cathode. However, in the case of the RGO/AgNP composite electrode, the wrapping of the RGO flakes around the AgNPs suppresses partially the Ag ⇄ Ag^+^ redox process.

It should be noted that the idea of the composite Ag/RGO electrode is somewhat different from that of the Ag/Pt electrode in standard IPMC. In the optimal operation mode, silver is only required to provide a sufficient electronic conductivity level, without creating an extra electrochemical actuation. Such actuation, associated with the release of silver atoms into the electrolyte, should ultimately lead to the silver depletion in the interelectrode space and to degradation of the actuator [20].

On this basis, we made an attempt to calculate and experimentally optimize the amount of silver in order to achieve the required conductivity level but to minimize the release of silver into the electrolyte volume at the same time [32].

We started with the requirement that the method of introducing silver into graphene electrodes should be relatively simple (e.g., as compared with the laser processing that will be concerned below). In contrast to [32], we did not use the PVDF-HFP composition as a polymer and BMIMBF_4_ as a filler. Instead, a “classical” Nafion membrane filled with water was used, with the substitution of H^+^ ions by Cu^+^; i.e., we tried to combine CNT modified membrane with a Nafion water-filled membrane.

The mechanism of how the graphene film conductivity is improved upon the introduction of silver is likely to consist in condensation of the silver particles at the junctions between the graphene flakes and providing conductive bridges between them. In addition, the introduction of silver can provide an additional mechanism for increasing flexibility, which will be discussed below.

Assessment of the optimal concentration of silver in graphene should take into account that the silver bridges between the graphene flakes are formed in a series of parallel processes.

At the first, during the reduction of silver from aqueous solution of Ag(NO_3_)_2_ and subsequent saturation of the solution with silver, the silver cluster condensation occurs primarily at the joints of the graphene flakes and then in the space between the flakes.

At the same time, the graphene in the flakes is reduced. Early in the process, the functional groups at the edges of the flakes and between them provide a sufficient space for silver atoms to join the silver clusters. Then, the space is gradually clinched to slow the condensation down. Another factor that interferes the condensation is a deformation of the graphene flakes placed between the silver clusters in the course of their growth (Figure 9).

Therefore, some silver concentration exists under which a secondary saturation of the cluster-graphene-aqueous solution system is to be awaited. The mean silver cluster size and the upper quantity of silver which should be inserted into the system are estimated in [32] at the basis of the new phase nucleation theory and graphene silver cluster system simulation in the HyperChem quantum chemistry package.

The estimation for the cluster mean size is
(1)$$R={\left(hL^{2}\left[\left(N_{\mathrm{flake}}-1\right)/2\right]/2\right)}^{1/3}$$
where *N*_flake_ is the graphene flake number in the intercluster space; 2*L* = *w* is the graphene flake size. Then, for a commercial GO flake of *w* = 3 μm size, the cluster radius *R* = 73 nm, which is a value correspondent with the mean cluster radius in SEM and TEM images in [33].

With the value of *R*, it is easy to estimate the total amount of silver in all the clusters. Assuming the density of particles in clusters to be equal to the density of solid silver, *N*_flake_ = 3, the C−C bond length in graphene *a*_CC_ = 1.42 Å, and the carbon atom mass *m*_C_ = 2.04 × 10^−23^ g, we obtain m_Ag_/m_G_ ≈ 1.

Besides that, the estimate shows that the required proportion of silver in the combined electrodes decreases with the flake size increase: at a size of 10 nm, it decreases by an order of magnitude.

The manufacturing technology of the RGO/Ag hybrid electrode and actuator as a whole was described by us in [32,34].

Figure 10 shows the conductivity dependence on the silver-to-graphene oxide mass ratio in the suspension, by evaporation of which the composite electrode was obtained. The electrode film is formed on the glass surface. It is seen that in the electrode from a suspension corresponding to the estimate (*m*_Ag_/*m*_G_ ≈ 1), the conductivity is only several times less than the possible maximum. We can take for this value *m*_Ag_/*m*_G_ ≈ 20–25, obtained in [32]. In our experiments, it is ~400 S/cm. The discrepancy in a factor ~2 in comparison with [31] can be attributed to the starting material quality and the specifics of its preparation.

Figure 11 shows the displacement amplitude dependencies on the frequency for a Nafion membrane with Pt electrodes (curve 1) and for a Nafion membrane with composite Ag/graphene electrodes at the same excess silver content as in [31] (curve 2). The voltage amplitude is 1 V. The results of [31] itself are shown in curve 3. It can be seen that curves 1–3 differ by no more than several times, depending on the frequency. The mechanical resonance was observed in [31] at lower frequency than in our work, which is a fact is associated with a small difference in the membrane thickness. However, with an optimized silver content, the characteristic displacement of the membrane (at a frequency of 1 Hz and the same voltage of 1 V) is slightly higher than in curves 1–3.

One of hypothetical mechanisms (in addition to the silver ions non-outcome into the electrolyte), which determines an increase in the system resource when using graphene, is the membrane flexibility increase. We made no attempt to specifically improve the adhesion of the electrode material to the membrane body. Nevertheless, it is known that in the course of GO reduction, the functional groups that remain on graphene are not uniformly distributed over its surface but are localized at certain places, creating a superstructure [35,36].

The areas where the functional groups are aggregated, have better flexibility than more rigid graphene flakes, and can provide better flexibility to the structure as a whole. It is natural to think that such reasoning should apply primarily to the GO with silver deposited on the glass surface in an ordered manner, e.g., with using the Langmuir–Blodgett technique. The same technique should optimally suit the structure considered in the model under study. Greater tightness of the electrodes should provide a longer lifetime of the system.

Another option for increasing both the graphene actuator efficiency and its lifetime is to ensure a stable adhesion between the graphene electrode and the electrolyte framework. In the actuator based on the ionic polymer-graphene composite (IPGC) [37] and characterized as extremely long-term durable, a hydrophobic graphene paper reduced (i.e., asymmetrically cleaned) by laser (hereinafter referred to as LrGO paper) was used. This paper consists of flexible, conductive outer surfaces (crack-free and hydrophobic) and laser-cleaned inner surfaces. Laser processing makes the membrane impervious to liquid, stops ion leakage during electrostimulation, and creates a strong bond at the ionic polymer-LrGO paper interface.

On the contrary, ions easily pass through microcracks in platinum, which arise in the electrodes of classical metal-film actuator, so that these electrodes are sufficiently permeable to water (Figure 12).

The production of the IPGC actuator includes vacuum filtration and laser scribing procedures. The GO paper is obtained by vacuum filtration of the GO suspension, and then, the paper is reduced with hydrogen iodide (HI) (Figure 13). Due to the iodine doping, the RGO paper of 5-mm thick has high conductivity (315 S/cm) and good flexibility.

Since the inner surface of the laser-cleaned GO is responsible for interaction with the electrolyte, it must be very rough. The properties of this surface were researched in isolated glovebox. The key property under study is wettability, characterized by the contact angle. It can be seen that, in LrGO paper, wetting changes to almost complete non-wetting. X-ray spectroscopy data (Figure 14) indicate physical reasons for this behavior.

Non-permeability of the membrane explains the fact that the water loss rate in the RGO paper is 0.0046 g/h, which is 10 times less than in the GO paper (Figure 15). At the same time, it is clear that IL cannot pass through the membrane since the size of its molecules is much larger than that of the water molecules. On the contrary, water can pass through RGO paper due to capillary effects in the GO: the functional groups of the GO in the paper “drag” water molecules.

The connection of the graphene electrode with the surface of the ion-conducting membrane, including the polymer framework and the electrolyte, providing a long-term adhesion and discussed above, is achieved thermomechanically. Such a connection is reached by partial diffusion of IL into the lower layers of the graphene electrode under hot pressure conditions. The graphene film, which forms the electrode, is synthesized separately.

The optimum conditions of this synthesis include a large number of parameters. Ideally, the film should include well-oriented graphene layers able to move freely parallel to each other. At the same time, this film, synthesized on a solid surface, should be sufficiently free to detach from it. As far as the authors know, the conditions for such synthesis were not researched since [38].

In [38], an ensemble of GO flakes was deposited from methanol solution onto the surface of quartz glass in Langmuir–Blodgett bath. Then, the reversibility of fusion and separation of flakes was investigated when the surface of the suspension was compressed or, conversely, expanded. A particular ensemble of GO flakes in [38] satisfied the requirement of reversibility of the film structure upon compression and expansion. However, [38] was focused on GO, and the determination of how reversibility is retained at different levels of the GO reduction has not been carried out. In addition, reversibility was highly dependent on the flake size dispersion.

Therefore, with a large size dispersion, when acts of collision of a small flake with a large one are frequent, the film tends to irreversible coagulation of flakes.

Anyway, manufacturing of the graphene film on a solid-state basis and its transfer to a polymer membrane is a complex technological procedure, which greatly increases the cost of the actuator as a whole. With additional laser treatment to improve the adhesion of the electrode and polymer membrane, an extra difficult procedure is added.

Unexpected enough, the effect similar to that induced by rather difficult laser processing was achieved when we used in the membrane the polymer, somewhat different from commonly used Nafion. This is the Russian analogue MF-4SK, which was actually proposed earlier than Nafion. The initial motive for choosing MF-4SK was laboriousness of the hot pressing method for combining the Nafion polymer membrane with the graphene. The study of the MF-4SK material in combination with traditional metal electrodes was described by us in [39].

In the paper [40] which followed [39], we combined the MF-4SK membrane with GO. The typical size of the GO flakes was 1–2 μm, up to 50% of the substance being in single-layer flakes. The oxygen content in the starting material, according to the manufacturers, was about 41%; the other components were sulfur (2–3%) and nitrogen (up to 1%).

To reduce the graphene from the GO, two methods of chemical reduction were used: in a water bath and in an oil bath. The details of the technology are presented in [40].

As the MF-4SK material had not been previously studied for the actuation problems, we started with pouring GO from dispersion as the simplest technique for covering a surface with the graphene. Previously, by a similar method, we failed to deposit the RGO on Nafion: the adhesion of the resulting film was too low. At the same time, the study of the combination of MF-4SK and metal electrodes revealed that in a number of parameters MF-4SK differs from Nafion significantly.

The MF-4SK membrane with a dry thickness of 120 ± 15 μm was pre-soaked in IL for several days. Then, a section on the membrane that seemed optimal was selected, and the partially reduced GO was drop-casted onto the membrane. It is essential that the entire area of the membrane was treated before cutting it into sections of individual actuators of a smaller area. After drying, a repeated coating was performed to ensure the required uniformity of the electrode film. Then the drop-casting procedure was repeated on the second side of the membrane.

Figure 16 shows the frequency dependencies of the actuator displacement amplitude when a sinusoidal voltage of 1 V is applied to the electrodes (curve 1) in comparison with the same curves for several modes with flexible metal electrodes. Curves 2,3 correspond to the results of [39]. The experimental technique did not differ from that described in [39] and in our earlier work [5].

It is seen that the actuation curve that corresponds the graphene electrode retains its amplitude almost constant in a very wide frequency range from 2–3 to about 12 Hz, in contrast to the IPMC dependencies, which drop drastically to zero with increasing frequency and appear again only at relatively high frequencies of the mechanical resonance. The curve for the Pt/MF-4SK sample with a thickness of 290 μm, not shown in Figure 16, almost merges with curve 2 at low frequencies, to appear only above a frequency of 70 Hz.

The dependence of the displacement amplitude on the applied voltage at low frequency of 1 Hz (Figure 17) also differs significantly from that for actuators with flexible metal electrodes. First of all, it is obviously linear within a wide voltage range. Accordingly, the control voltage calibration in a particular device is simplified.

Figure 16 shows that, when using the graphene electrodes, the actuation amplitude in the low frequency region is lower than in the case of traditional Pt electrodes. This is obviously correlated with the fact that, in the drop-casting method, the surface resistance of the electrode film did not decrease lower than 1.5–2.0 kOhm on the MF-4SK membrane and 3–4 kOhm on the Nafion membrane. Meanwhile, the platinum layer resistance in good membranes produced by standard technique has the order of 100–200 Ohm.

Except actuation characteristics, this simple method enables better uniformity and adhesion of the graphene layer to the polymer surface than to Nafion. The origin of this result is clearly seen from the analysis of the AFM profilograms of the MF-4SK polymer surface in [39]. The main result is an increase in both the average pore size and the dispersion of this size, which manifests itself in the roughness increase. When working with the graphene electrode, this roughness increase of the polymer should lead to an improvement in the graphene electrode adhesion to the polymer.

This result correlates well with the effect of improving the adhesion when the GO paper electrode is processed with a laser [37]. One can assume that this effect is partially appearing from additional recovery of the GO. However, it is the roughness increase that the authors of [37] focused on. Indeed, roughness reduces the “slippage” of the graphene paper along the polymer and forces the graphene to stretch together with this polymer.

In order to understand the features of how the MF-4SK/RGO structure operates, we turned to the known experimental trends of this material, which distinguish it from the Nafion. They can be summarized as follows [41].

The Nafion polymer chains have a slightly shorter length *l*_ch_ (9.3 nm versus 9.8 nm in the MF-4SK) and size dispersion (2.15 nm versus 2.9 nm in the MF-4SK), if the Gaussian distribution dispersion is understood as the size under which the proportion of chains decreases by the factor e^2^ ~ 10 times (5.0–13.6 nm in the Nafion and 4.0–15.6 in the MF-4SK) [41].

The mean size and the size dispersion of the Nafion’s water-filled pores in the water saturated polymer are ~5 nm and ~1.0 nm (5.0 ± 2.0 nm), respectively, and these values depend on the model accepted for the membrane structure. Anyway, the pores are connected by channels that ensure the permeability of the membrane for aqueous solutions, most of the pores being open. The water-filled pores in the MF-4SK have a larger scatter of sizes (5.0 ± 4.0 nm); herewith, the fraction of isolated pores in MF-4SK is much higher.

The model developed in [40] successfully interprets a number of experimental facts: the higher roughness of the MF-4SK surface (that provides it with good adhesion to the graphene film), the higher permeable pore size (that provides better actuation characteristic).

Except the polymer surface roughness, the reduction level of the GO in hydrazine plays a key role in the operation of the actuator, providing the required level of the negative functional groups substitution with amino groups is reached. In [40], the water bath processing in the most successful experiments was 1 h. Longer processing led to resistance increase although the activation value dispersion in several experiments was wide.

It should be noted that attempts to precipitate the aqueous dispersion of GO, which has already amino-modified (1 mg/mL), was unsuccessful. After the water bath, the aqueous dispersion of the reduced amino-modified GO was destroyed, and the graphene derivatives precipitated.

Considering the operation of actuators with graphene electrodes, it is necessary to mention an alternative class of flexible electrode materials that is competitive with the graphene in terms of price. One of these options is the PEDOT:PSS conductive polymer. In [42], we analyzed the operation of such actuators in comparison with the electrodes made of noble metal nanoparticles [43]. The operation modes in air and in water, both deionized and containing CuSO_4_ ions with significant ionic strength, were analyzed in detail.

The results of this research can be summarized as follows. Actuators with the PEDOT:PSS electrode have slight performance advantage at low electrode voltages compared to the noble metal actuators, especially in deionized environments. A similar detailed comparison with the graphene electrodes in the entire set of operation environments and modes has not been performed. However, no significant difference in the efficiency was observed. From the viewpoint of the cost of both the synthesis and the technology of combining electrodes with polymer membranes, the difference was also insignificant. Therefore, one can think that the choice of one or another electrode should be determined then by the characteristics of the environment and additional requirements (e.g., color).

To summarize all the above, a number of nanocarbon materials and conductive polymers ensure steady, long-term operation of the actuator both in air and in liquid medium as the electrode material. They also have sufficiently high amplitude of the actuator displacement and good frequency characteristics. At the same time, in all the works described, the disadvantage of the electrodes based on *sp*^2^-carbon is very low energy transduction efficiency (no more than 1%). This result is a sequence of the fact that the both actuation mechanisms involved in the carbon nanostructure (quantum and electrostatic stretching associated with the diffusion and the adsorption of ions) do not use the material’s own elastic properties.

These properties are realized in graphdiyne, which is a specific carbon form proposed by the authors of [44,45,46], as the most steady carbon allotrope with a high level of π-bonding.

## 5. Actuators with Graphdiyne Electrodes

Graphdiyne layers include hexagonal cells stabilized by binding carbyne-type chains (more precisely, by diacetylene groups –C≡C–C≡C–); i.e., each such layer includes *sp*- and *sp*^2^-hybridized carbon atoms. The planar lattice of chains forms a system of triangular pores and possesses by tunable electronic properties. The interaction between the layers is much weaker than the covalent interaction and is similar to interactions between the graphene layers.

Similar to individual graphene layers, the graphdiyne layers can form graphdiyne nanotubes by twisting from the edge of the “zigzag” or “armchair”-type nanoribbons [47]. Examples of the armchair graphdiyne nanotubes are shown in Figure 18.

Unlike “classical“ CNTs, graphdiyne tubes have porous surface and can be used for hydrogen energetics as fuel accumulators. The carbon bond lengths in the graphdiyne nanotubes are not the same due to different types of hybridization in the carbon rings and diacetylene bridges.

From the viewpoint of electronic properties, all the graphdiyne nanotubes are semiconductors with a band gap Δ*E*_g_ down to 1 eV. Thus, it is natural that they are easier to work than the classical CNTs. With an increase in the graphdiyne tube diameter, Δ*E*_g_ decreases.

Coming back to the porous 3D structure of the flat graphdiyne layers, it should be noted that the structure suggests the possibility of the ion diffusion either along the layers or between the layers with small barriers.

The bridges between the graphdiyne layer rings (i.e., between the hexagons) can easily form molecular scale alkene-alkyne complexes. This complexation is important for electronics, photovoltaics, the energy storage, and catalysis [48,49].

Since the graphdiyne is a less known material than the nanotubes and the graphene, we will make some comments on its synthesis technique. Graphdiyne is synthesized on the surface of copper through cross-coupling reactions with hexaethynylbenzene, as shown in Figure 19. The most difficult chemical to obtain in this synthesis is hexakis[(trimethylsilyl)ethynyl]benzene, which was dissolved in tetrahydrofuran and ammonium chloride in an ice bath under nitrogen atmosphere and kept for the required time. Further procedures are described in [44,45,46] and do not look more complicated than the synthesis operations of acceptable quality GO.

In the SEM image, graphdiyne is composed of nanoparticles that aggregate to form a porous hierarchical structure [45]. In contrast to the graphene, which has a laminar morphology, each of the graphdiyne nanoparticles can be characterized as a lamellar structure with interlayer distance of 0.37 nm.

When using in the actuators, the graphdiyne is interesting from the viewpoint of the tensions caused by molecular structural changes. This problem is put in [49], whose logic we will follow.

The assembly diagram and the structure of actuators with graphdiyne electrodes are shown in Figure 20. The assembly is based on the hot pressing technique of the graphdiyne film into a solid composite electrolyte filled with IL.

Herewith, the SEM shows that the adhesion to the electrolyte is very good.

Comparative cyclic voltammograms (CV curves) for electrodes with the CNT, graphene, and graphdiyne, which formed the actuators of the same structure, are shown in Figure 21.

One can see that the actuator CV curves measured on the PVDF/EMIMBF_4_ polyelectrolyte with an electrochemical window of 2.5 V and a scanning rate of 10–100 mV/s have a regular rectangular shape, i.e., exhibit good capacitive behavior. The specific capacity of the actuator with graphdiyne electrodes was 237 F/g at a current density of 1 A/g and a scanning rate of 10 mV/s, which is a value almost five times higher than that of the actuator with CNT electrodes.

It is important that the specific capacity of the actuator remains at a level of 200 F/g, which is inaccessible for the graphene and the CNTs as electrode material, even at a scan rate of 100 mV/s (Figure 21d). The ability of the actuator with graphdiyne electrodes to accumulate energy stems from particular triangular shape of the nanopores, from large number of highly active alkyne sites and from porous hierarchical structure with sufficiently high specific surface area (865 m^2^/g).

The triangular pores and the porosity of the structure allow the ion migration during the charge/discharge process, whereas high specific surface makes the electrode act as an ion reservoir. The combination of these factors guarantees good energy storage.

The schematic of the ion movement and bimorph actuator bending is shown in Figure 22.

Initially, the cations and anions migrate to the cathode and anode, respectively. Then the cations interact with the graphdiyne structure and transform the alkyne complexes into the alkene ones to change the lengths in the covalent bond network in the cathode. There is no such transformation at the anode, where the anions enter. Unbalanced stretching of the opposite electrodes leads to stretching the cantilever bimorph structure as a whole.

With the frequency increase, the actuator displacement amplitude decreases, just as it should be (in Figure 23, the voltage amplitude is 2.5 V). However, the advantage of actuators with the graphdiyne electrodes over that with the CNT- and graphene-based electrodes remains valid at all frequencies.

It is seen that an elongation of 0.07% is provided for the graphdiyne actuator even at a frequency of 30 Hz, while the actuators with the CNT- and graphene-based electrodes practically cease to work already at 10 Hz.

The other fundamental technical results are as follows. The energy density (the inset in Figure 24) in the actuator with graphdiyne electrodes was 12 kJ/m^3^, which is a value comparable to that in the mammalian musculature skeleton (~85 kJ/m^3^). The Young’s modulus, determined from the tension-load curves in Figure 24, was 420 MPa. The mechanical stress which developed during the transformation of the alkyne complex into the alkene one was 3.11 MPa, i.e., much higher than that of human skeletal muscles (0.3 MPa).

The blocking force of the graphdiyne actuator at a voltage amplitude of 2.5 V and at a frequency of 0.1 Hz was 3.37 mN; i.e., it exceeded the blocking force of the actuator with the CNT electrodes (1.38 mN) and the graphene electrodes (1.92 mN) several times.

The most significant progress has been made in the energy transduction efficiency. It reached 6%, which is a factor several times higher than that of any actuator material such as piezoceramics, shape memory melts, and magnetostrictive materials.

The actuator operated successfully within the range of 0.1–30 Hz for up to 100,000 cycles.

The mechanism of how the alkyne complex transforms into the alkene one is described in [49] on the example of changing one quasi-plane layer and one structural unit (Figure 25).

Actuated by electric voltage, the ions surrounding the graphdiyne site (EMIM cations) are adsorbed onto this site and then coordinated with the alkyne bonds. Due to the presence of an unbound orbital, alkyne bonds are converted into alkene bonds, and the bond length changes. This bond length transformation leads to a change in the size of the covalent bond network. The transition process is reversible and controlled by an external field.

The theoretical value of the graphdiyne stretch during the complexation can be estimated from the change in the bond length, which is quite successfully simulated by various applied packages of quantum chemistry. It turned out that the elongation reaches 17%.

The energy storage mechanism in the graphdiyne-based electrodes is that numerous alkyne sites of high ionic complexation activity enhance the graphdiyne binding with cations and increase the charge density of the electrodes during both the charge and discharge processes.

## 6. Conclusions

The technology of the actuator electrodes based on nanocarbon is a promising area in the development of actuators. It allows avoiding the use of expensive noble metals and at the same time partially overcomes the problems of their use. For the electrodes made of graphene, which is the most promising and at the same time the most available nanocarbon material, both the materials of flexible polymer and the media that fills it, giving high response to external signal, have been found.

To our opinion, such material, principally corresponding to the Nafion, is its Russian analogue MF-4SK with water filling and substitution of water cations by Cu^2+^ cations. Filling the polymer framework with an ionic liquid instead of the aqueous electrolyte provides weaker response than the aqueous electrolyte, which creates an extensive hydrated shell around the cations.

On the other hand, the MF-4SK material provides good graphene adhesion to the flexible polymer and quite high conductivity of the electrode film even if it is applied from the partially reduced graphene oxide by the simplest method (drop-casting). The functionalization of the graphene with amino groups provides its electrostatic attraction to the negative groups on the Nafion surface. As a high level of this functionalization inevitably decreases the film electrical conductivity, the experiment with an acceptable degree of reduction can be considered a non-trivial result. Further progress in this case could be reached on the way of varying this functionalization. To a lesser extent, this applies to the traditional interelectrode membrane material, Nafion.

A number of other methods are known to meet the conflicting properties demanded of actuators. These are based on hot clamping technology and are complemented by various methods (e.g., laser processing) to avoid electrode cracking and drying and to guarantee a long service life. The technology of hot clamping is based on the partial penetration of the polymer deep into the graphene film, which is synthesized on a separate solid surface, then separation from this surface, and transferring to the polymer. Our experience shows that this method, when applied to a water-filled polymer like Nafion, is very challenging.

In the case of polymers filled with ionic liquids, it is more successful, but the response of the actuator thus obtained is significantly weaker. Therefore, progress in such systems can be achieved by improving the quality of the film, which provides the maximum possible conductivity.

By improving the quality, it seems to increase the size of the flakes and reduce their dispersion in size.

In connection with the above, the problem of a clear classification of the graphene material and its properties is urgent. This requirement has been put forward as one of the main ones at graphene conferences of recent years.

If we consider that the issue of obtaining sufficiently cheap graphene and its classification has been resolved, graphene can be used to obtain actuators that are cheap to manufacture.

In this case, it seems that methods of obtaining actuators for operation in air and in a liquid medium should be separated.

The use of graphdiyne-based actuators represents a qualitatively higher level of operational efficiency. At the same time, the method of obtaining the material itself is currently expensive. It can be assumed that the ratio between the use of these two materials can be similar to the use of monocrystalline silicon for obtaining high-efficiency solar cells and polycrystalline silicon for cheap cells with significantly lower efficiency. A further perspective may be the use of hypothetical carbon forms, similar to graphdiyne, but with an even stronger mechanical response to a change in the state of the connecting acetylene chains in them. Such forms can be graphiyne, super diamond, and some others.

## Figures and Tables

**Figure 1 polymers-13-04137-f001:**
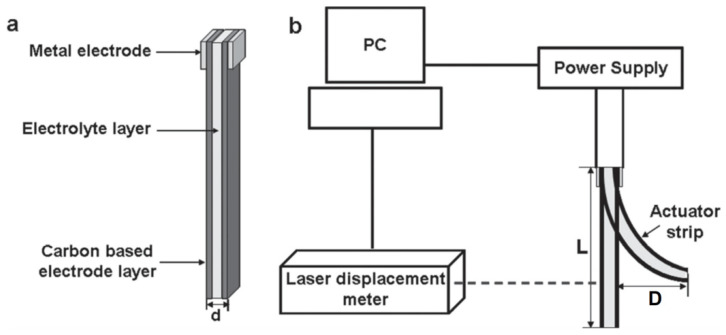
IPMC actuator: (**a**) Schematic of the structure; (**b**) schematic of the measurement. Reproduced from [3] with permission from John Wiley and Sons. Copyright 2013.

**Figure 2 polymers-13-04137-f002:**
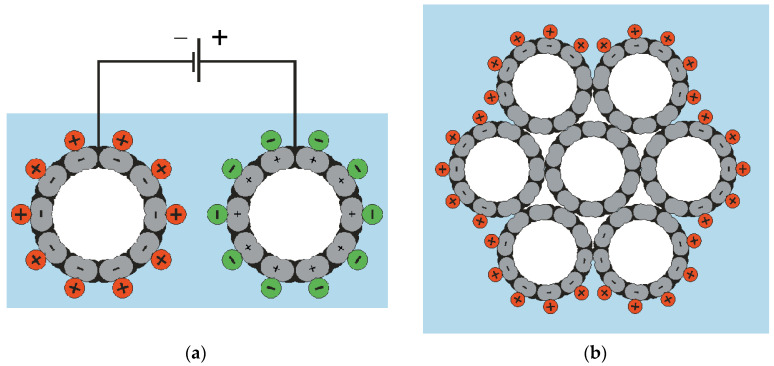
Electrostatic diagram of how one of the nanotube electrodes elongates and the other electrode constricts: (**a**) both electrodes are represented as individual nanotubes; (**b**) the electrode is represented as a bundle of nanotubes. Reproduced from [13] with permission from the American Association for the Advancement of Science. Copyright 1999.

**Figure 3 polymers-13-04137-f003:**
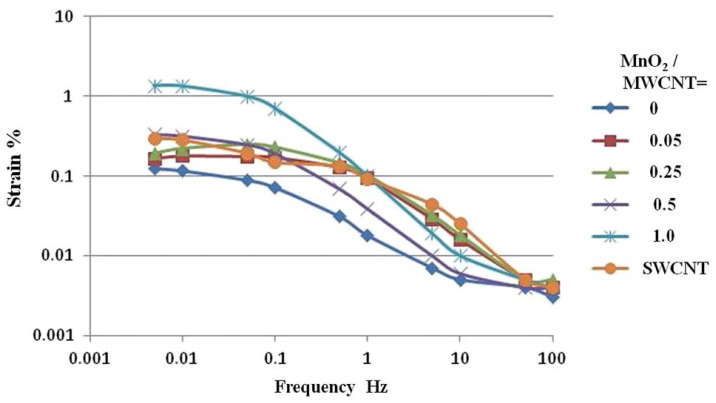
Actuator stretching, recalculated from the peak-to-peak displacement of the actuator with the gel electrode of “polymer/CNT/EMIM[CF_3_SO_3_]” and different concentrations of MnO_2_ in relation to CNTs (sets of points MnO_2_/MWCNT = 0, MnO_2_/MWCNT = 1 are selected). Reproduced from [26] with permission from the Royal Society of Chemistry. Copyright 2016.

**Figure 4 polymers-13-04137-f004:**
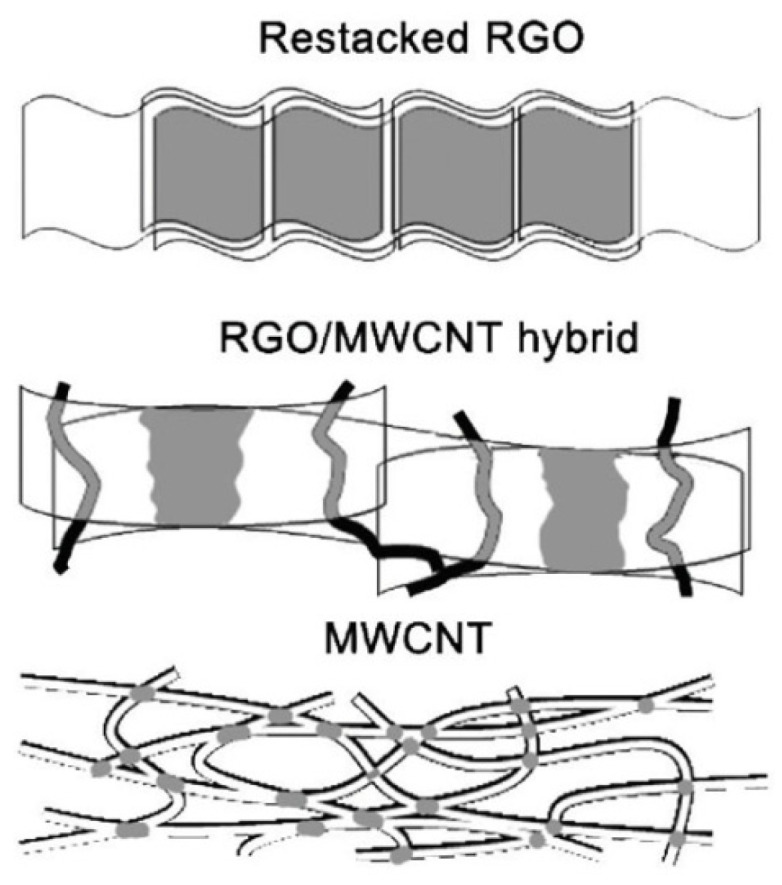
Schematic of how the restacking factor of the graphene layers is blocked when the CNTs are inserted into the composite electrode. Reproduced from [30] with permission from John Wiley and Sons. Copyright 2012.

**Figure 5 polymers-13-04137-f005:**
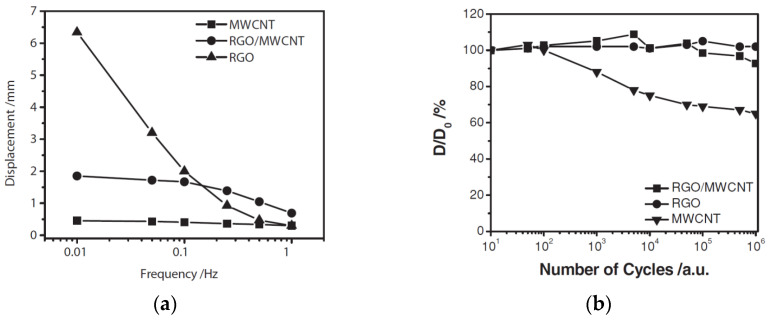
Characteristics of the actuators with electrodes based on MWCNTs, graphene (RGO), and their composite (MWCNT/RGO): (**a**) frequency dependencies of the displacement amplitude; (**b**) operational stability depending on the number of cycles. Reproduced from [30] with permission from John Wiley and Sons. Copyright 2012.

**Figure 6 polymers-13-04137-f006:**
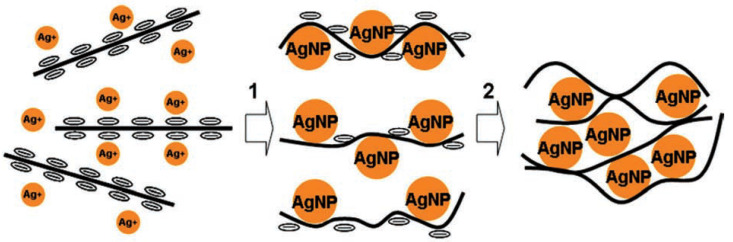
Schematic diagram of how the AgNPs are formed in the space between the graphene flakes during the reduction. Reproduced from [31] with permission from John Wiley and Sons. Copyright 2012.

**Figure 7 polymers-13-04137-f007:**
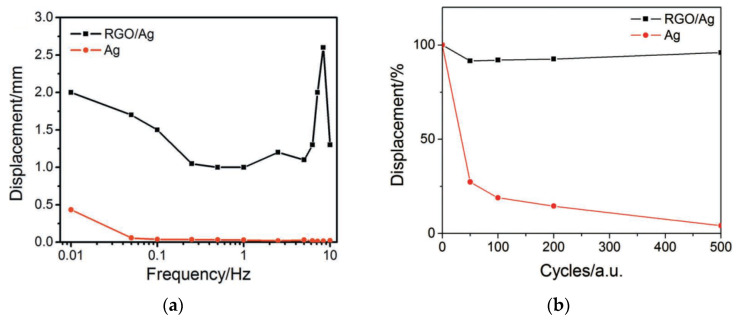
Dependencies of the displacement amplitude of the actuator with combined and pure silver electrodes (**a**) on the frequency and (**b**) on operation time. Reproduced from [31] with permission from John Wiley and Sons. Copyright 2012.

**Figure 8 polymers-13-04137-f008:**
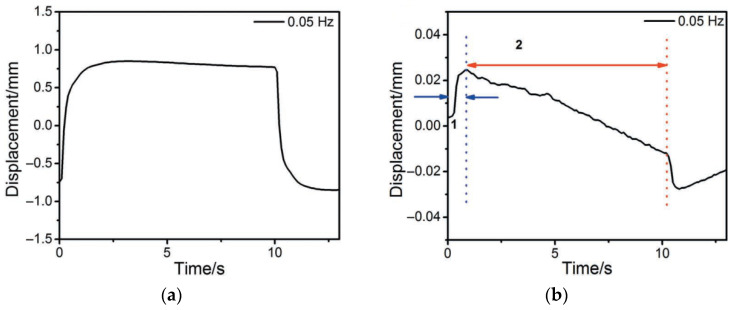
Actuation curves for meander-type signals of the actuator with electrodes from (**a**) RGO/Ag and (**b**) Ag. The frequency and the amplitude of stimulation are 0.05 Hz and 1 V, respectively. Reproduced from [31] with permission from John Wiley and Sons. Copyright 2012.

**Figure 9 polymers-13-04137-f009:**
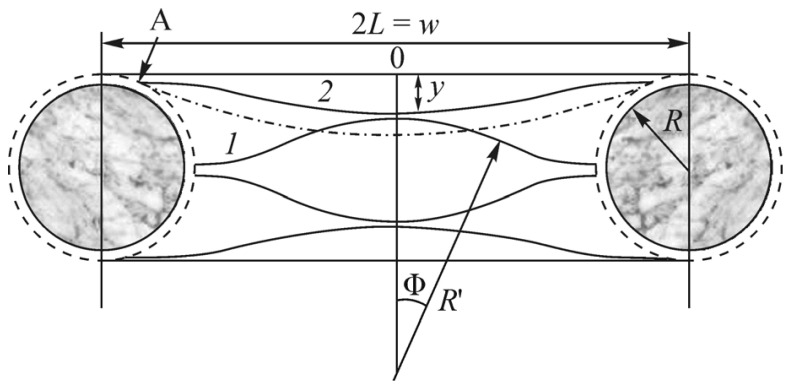
Schematic of how the graphene flakes of the same size, 2*L* = *w,* are deformed being clenched in the space between silver clusters of radius *R* when the clusters grow. Reproduced from [32] with permission from Springer Nature. Copyright 2018.

**Figure 10 polymers-13-04137-f010:**
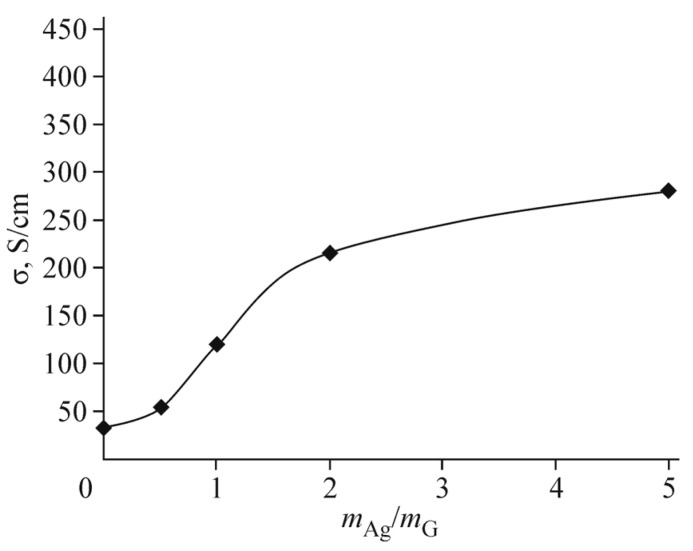
Conductivity of the insulated Ag/RGO composite electrode versus the silver/graphene ratio. Reproduced from [32] with permission from Springer Nature. Copyright 2018.

**Figure 11 polymers-13-04137-f011:**
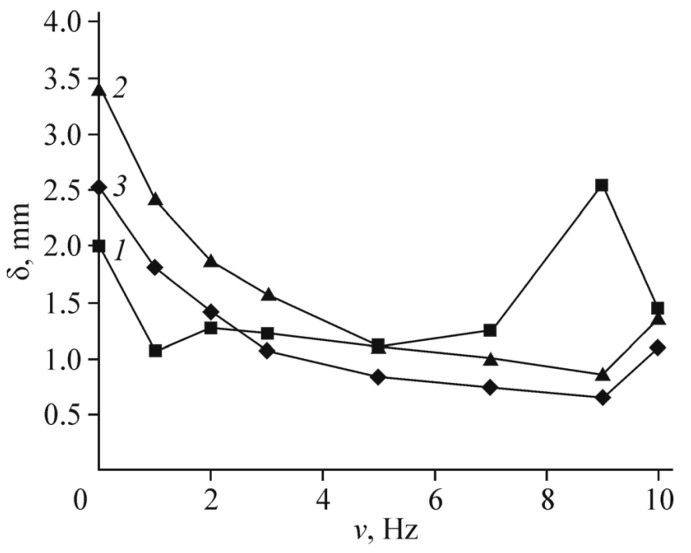
Frequency dependencies of the actuator displacement amplitude for Nafion membranes with (1) Pt electrodes and (2) composite Ag/graphene electrodes with the same silver content as in [31]; (3) results of [31]. The voltage amplitude is 1 V. Reproduced from [32] with permission from Springer Nature. Copyright 2018.

**Figure 12 polymers-13-04137-f012:**
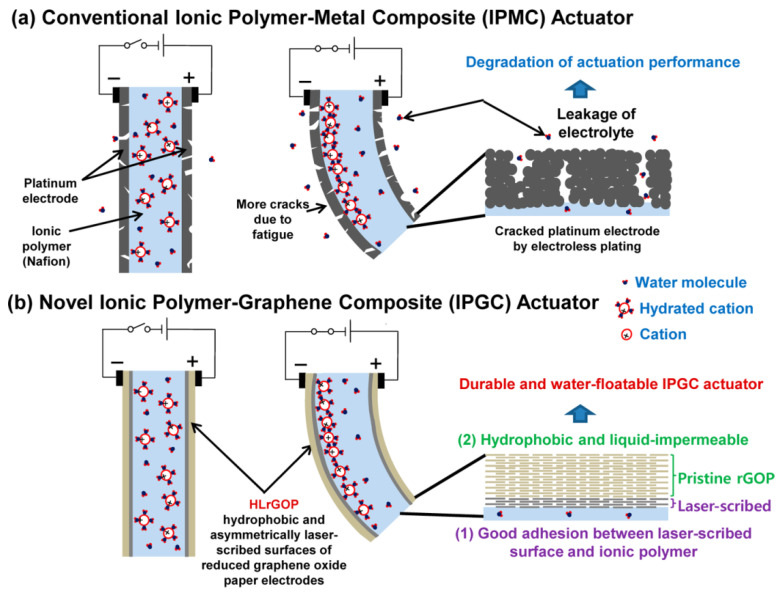
Actuators with different electrodes: (**a**) IPMC actuator, permeable for liquid; (**b**) IPGC actuator, allowing long-term operation. Reproduced from [37] with permission from American Chemical Society. Copyright 2014.

**Figure 13 polymers-13-04137-f013:**
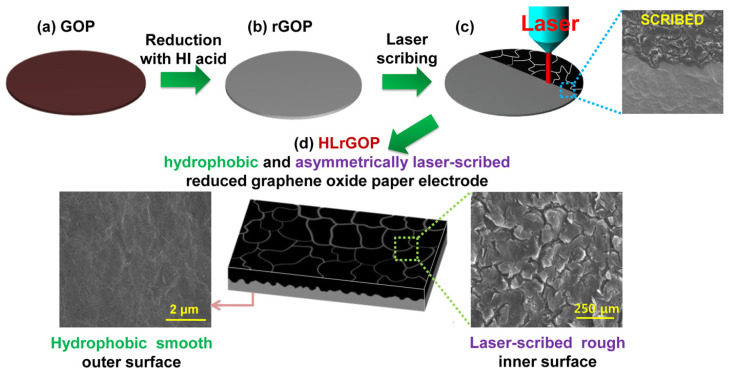
Synthesis of the LrGO paper electrode with hydrophobic and asymmetric laser-cleaned surfaces: (**a**) GO paper (GOP); (**b**) GO paper, reduced with HI (rGOP); (**c**,**d**) laser-cleaned RGO paper (LrGOP). Reproduced from [37] with permission from American Chemical Society. Copyright 2014.

**Figure 14 polymers-13-04137-f014:**
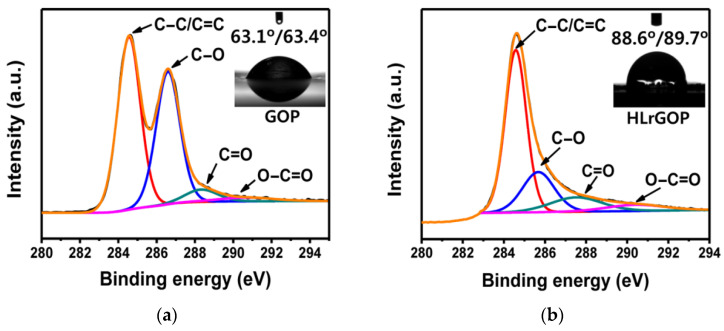
X-ray spectroscopic data showing C1s peaks and contact angle corresponding to (**a**) wettability of the GO paper with water and (**b**) non-wettability of the GO paper reduced by laser. Reproduced from [37] with permission from American Chemical Society. Copyright 2014.

**Figure 15 polymers-13-04137-f015:**
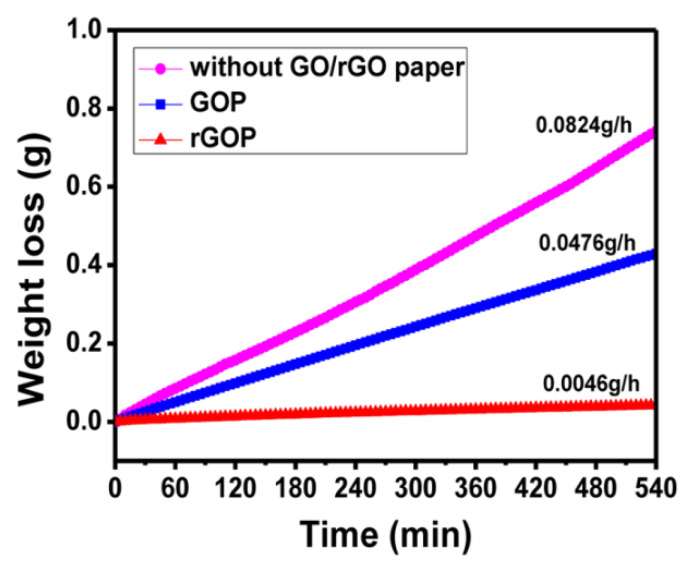
Wettability of the GO paper and RGO paper and its effect on the actuator operation time. Reproduced from [37] with permission from American Chemical Society. Copyright 2014.

**Figure 16 polymers-13-04137-f016:**
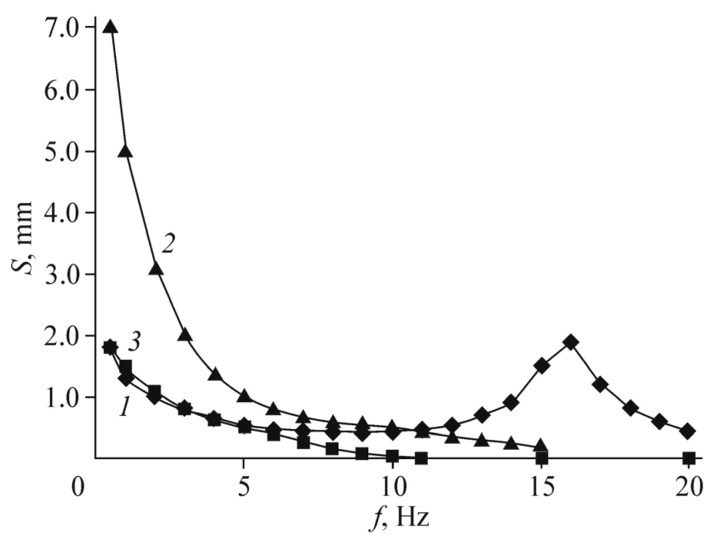
Frequency dependencies of the actuator displacement amplitude against sinusoidal applied voltage (the amplitude is 1 V) for (1) graphene electrodes on MF-4SK and (2, 3) flexible platinum electrodes researched in [39]: (2) Pt/Nafion (125-μm thick); (3) Pt/MF-4SK (120-μm thick). Reproduced from [40] with permission from Springer Nature. Copyright 2020.

**Figure 17 polymers-13-04137-f017:**
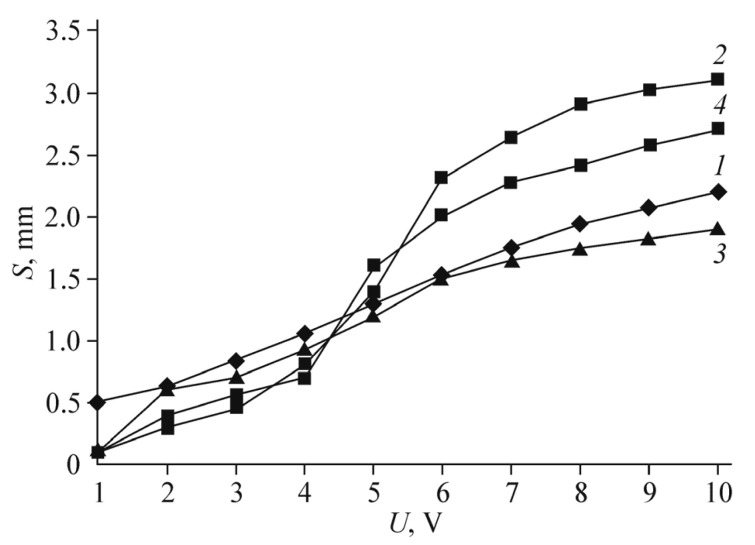
Dependencies of the actuator displacement amplitude against sinusoidal applied voltage (the frequency is 1 Hz) for the electrode-membrane combinations: (1) graphene/MF-4SK; (2) Pt/Nafion (125-μm thick); (3) Pt/MF-4SK (120-μm thick); (4) Pt/MF-4SK (290-μm thick). Reproduced from [40] with permission from Springer Nature. Copyright 2020.

**Figure 18 polymers-13-04137-f018:**
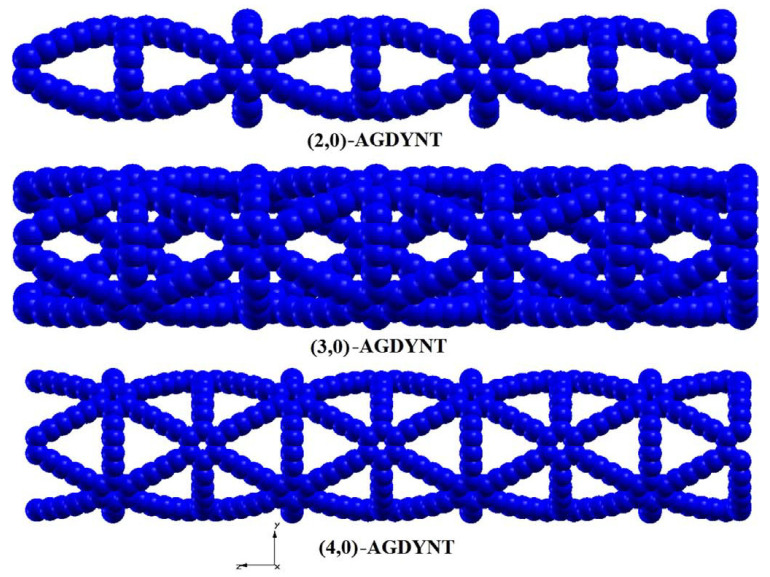
Examples of optimized armchair graphdiyne nanotubes (AGDYNTs). Reproduced from [47] with permission from Elsevier. Copyright 2016.

**Figure 19 polymers-13-04137-f019:**
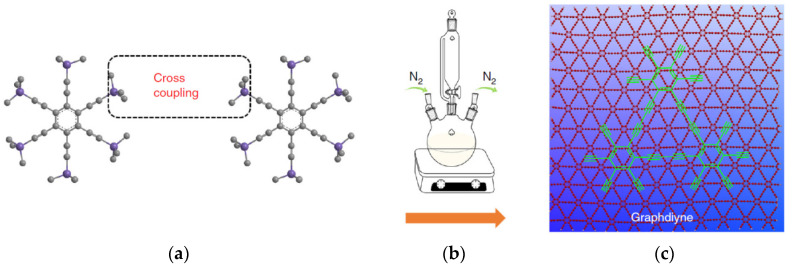
Formation of the flat graphdiyne network: (**a**) synthesis schematic; (**b**) principal setup of the synthesis; (**c**) ideal graphdiyne layer structure. Reproduced from [49] with permission from Springer Nature. Copyright 2018.

**Figure 20 polymers-13-04137-f020:**
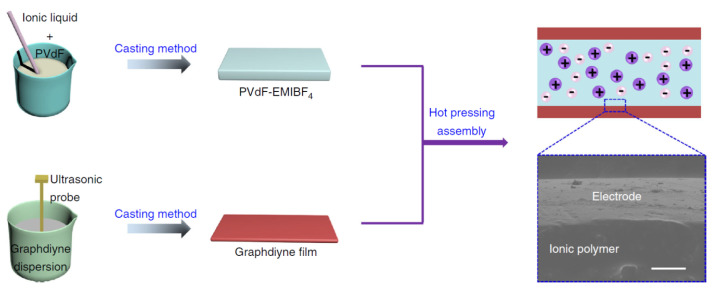
Schematic for the manufacture of graphdiyne electrodes, an electrolyte-containing membrane, and an actuator membrane as a whole. Reproduced from [49] with permission from Springer Nature. Copyright 2018.

**Figure 21 polymers-13-04137-f021:**
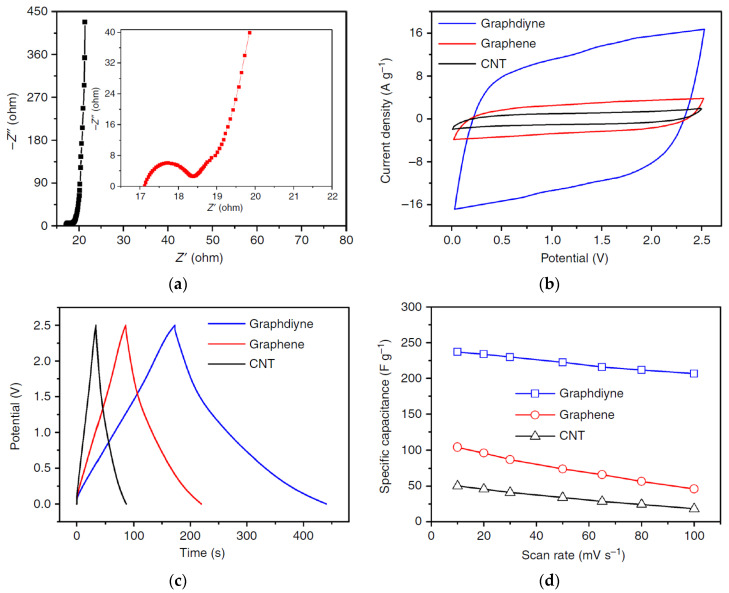
Electrochemical properties of the graphdiyne actuator: (**a**) Nyquist curves (an increased range of 0–22 Ohm is shown on the inset); (**b**) comparative CV curves for electrodes based on CNT, graphene, and graphdiyne forming actuators of the same structure (the scan rate is 100 mV/s); (**c**) charge/discharge curves for actuators with electrodes from CNT, graphene, and graphdiyne; (**d**) specific capacity of the actuators as a function of current density. Reproduced from [49] with permission from Springer Nature. Copyright 2018.

**Figure 22 polymers-13-04137-f022:**
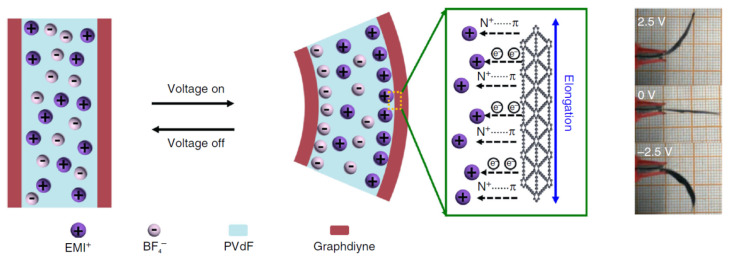
Schematic of the actuation mechanism in actuators with the graphdiyne electrodes. The image shows the actuator displacement at a voltage amplitude of 2.5 V and a frequency of 0.1 Hz. Reproduced from [49] with permission from Springer Nature. Copyright 2018.

**Figure 23 polymers-13-04137-f023:**
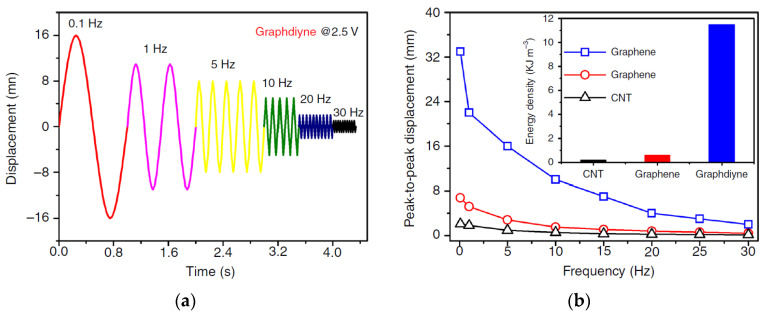
Efficiency of the actuator with the graphdiyne electrodes: (**a**) sinusoidal displacement on time at different frequencies; (**b**) peak-to-peak displacement as a function of frequency for the electrodes of three types (the insert illustrates the comparison of energy densities). Reproduced from [49] with permission from Springer Nature. Copyright 2018.

**Figure 24 polymers-13-04137-f024:**
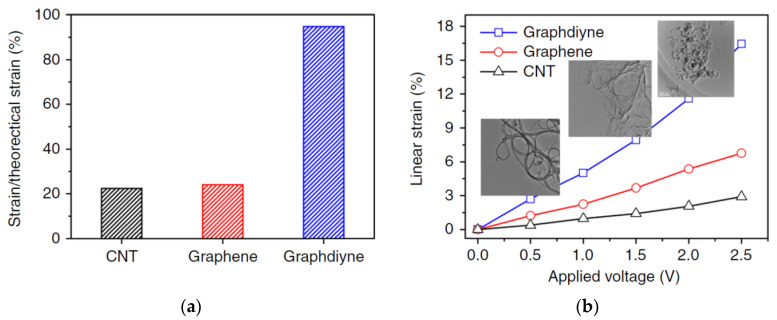
Linear stretch of the graphdiyne-, graphene-, and CNT-films (**a**) in relation to the theoretically calculated values at a fixed voltage of 2.5 V and (**b**) depending on the applied voltage. The inset shows the TEM images for these three materials. Reproduced from [49] with permission from Springer Nature. Copyright 2018.

**Figure 25 polymers-13-04137-f025:**
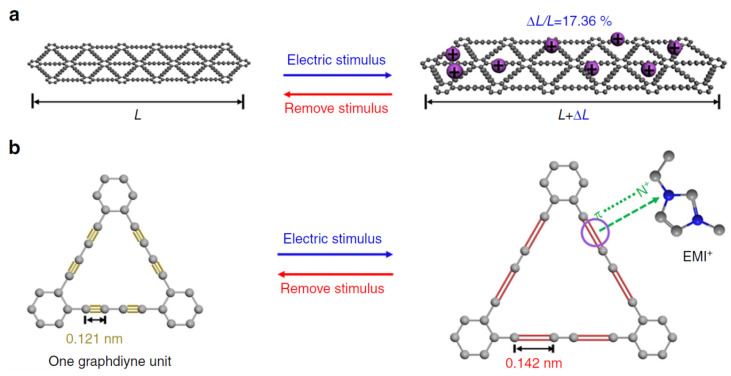
Actuation mechanism of the graphdiyne material: (**a**) schematic of the actuation tension of graphdiyne when interacting with an electrical stimulus; (**b**) mechanism of the transition between the alkene and alkyne complexes in a separate graphdiyne node. Reproduced from [49] with permission from Springer Nature. Copyright 2018.

## Data Availability

Not applicable.

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
