# Peer review of "Ionic EAP Actuators with Electrodes Based on Carbon Nanomaterials"

_polymers, 2021, doi:10.3390/polym13234137_

Round 1

Reviewer 1 Report

The manuscript « Ionic EAP Actuators with Electrodes Based on Carbon Nanomaterials» is an extensive logically complete interesting overview. The topics covered are relevant and in demand. The presented data are supported by bright informative figures. Previously obtained results are qualitatively summarized and analyzed. The review describes in sufficient detail the types of electrodes of actuators, electroactive polymers – membranes, production methods and fields of application. A detailed conclusion has been prepared. However, for an article on the topic of actuators in the journal Polymers, it is necessary to provide information on the nature of the effect on which the actuators are based and what variants of the mechanisms causing deformation may be, including an emphasis on the role of the polymer (membrane). The results of the graphs Fig. 3, 5, 7, 11, 16, 23 clearly indicate the significant contribution of diffusion of charge carriers, perhaps the authors can supplement the manuscript with a generalizing paragraph on the effect of diffusion, including in the polymer (membrane). In the annotation, lines 29-31 are unrelated to the previous text, despite being fully consistent with the review text. Changes need to be made: remove or add one more sentence for clarification. I recommend adding the genre of the review in the title of the article. Accept after minor revision.

Author Response

Dear Reviewer,

Thank you for your comments!

Our response is attached below.

Reviewer 2 Report

This is a very well written and extensive review. The focus is novel and timely. In general, the authors used a clear narrative style and organized a paper in such a manner that provides all necessary information concerning carbon actuators in composite materials (with polyelectrolytes, ionic liquids. etc.),  based on the author's previous work. Overall, the manuscript provides all comprehensive information for scientists and other readers interested in the composite materials that act as actuators.

The manuscript is suitable to publish.

Author Response

Dear Reviewer,

Thank you for your decision!

Sincerely, Ivan K. Khmelnitskiy
